# The portrayal of antimicrobial resistance in Bangladeshi newspapers during 2010–2021: Toward understanding the narrative

Tahmidul Haque[1]*, Syed Hassan Imtiaz[1], Md. Imran Hossain[1], Sazzad Hossain Khan[1], Md. Mahfuj Alam[2], Zahidul Alam[3], S. M. Rokonuzzaman[1], Orindom Shing Pulock[1], Susmita Dey Pinky[1], Ataul Karim Arbi[4], Haroon Bin Murshid[1], Nusrat Homaira[5], Md. Zakiul Hassan[1]

1 International Centre for Diarrheal Disease Research, Bangladesh (icddr,b), Dhaka, Bangladesh, 2 Department of Epidemiology, University of Massachusetts Amherst, Amherst, Massachusetts, United States of America, 3 Department of Surgery, Sir Salimullah Medical College and Mitford Hospital, Dhaka, Bangladesh, 4 Directorate General of Health Services (DGHS), Ministry of Health and Family Welfare (MOHFW), Government of the People's Republic of Bangladesh, Dhaka, Bangladesh, 5 Discipline of Pediatrics, University of New South Wales (UNSW), Sydney, Australia

* aunitahmid83@gmail.com

**Data Availability Statement:** All relevant data are within the paper.

## Abstract

### Background

Antimicrobial resistance (AMR) is a major global public health crisis and around the last decade, newspapers were one of the main sources of public dissemination of information for so. This study highlights how Bangladeshi mainstream newspapers represented AMR-related news and how they created the narrative of AMR in Bangladesh.

### Methods

We conducted both quantitative and qualitative content analysis on 275 AMR-related news articles published in the twelve highest circulated dailies (January 2010 to September 2021). We divided the articles into report, opinion, and editorials and analyzed how their contents built the narrative of AMR in Bangladesh.

### Results

Bangladeshi newspapers reported misuse of antibiotics by the consumers the most (32.2%), followed by selling without prescriptions (29%), and over-prescription by the health providers (26.1%). There were hardly any news reports describing the impact of pharmaceutical companies in prescribing and selling antibiotics. Around 45% of the news articles were event-oriented. Moreover, they suggested inadequate recommendations to battle AMR.

### Conclusion

Valid, consistent, and reliable AMR news coverage can play a crucial role in creating mass awareness, making providers accountable, and supporting national action plan in mitigating

**Funding:** The authors received no specific funding for this work.

**Competing interests:** The authors have declared that no competing interests exist.

**Abbreviations:** AMR, Antimicrobial Resistance; GDP, Gross Domestic Product; SDG, Sustainable Development Goal; LMIC, Low and Middle-income Country; WHO, World Health Organization; SEARO, South-East Asian Regional Office; OTC, Over the counter; IPC, Infection prevention and control; WAAW, World antibiotic awareness week; MOHFW, Ministry of Health and Family Welfares; DGHS, Directorate General of Health Services; IEDCR, Institute of Epidemiology, Disease Control and Research; DGDA, Directorate General of Drug Administration; BSMMU, Bangabandhu Sheikh Mujib Medical University; icddr,b, International Centre for Diarrheal Disease Research, Bangladesh; TRP, Target Rating Point.

AMR threat. The Bangladeshi journalists interested in reporting AMR-issues should focus on disseminating more Bangla articles with scientific information, and reporting causes and recommendations responsibly.

## Background

The emergence of anti-microbial resistance (AMR) and subsequent health problems are increasingly being recognized as one of the major global public health threats [1, 2]. Although antibiotics have been one of the key inventions of medical science for treating life threatening infectious diseases, its unwarranted and inappropriate use have been the major drivers of emergence of AMR globally [1, 3]. Today, we run the risk of living in a "post-antibiotic" world where the choice of antibiotics for treatment has been shrinking gradually and thus, has created a complex situation for the healthcare system [4, 5]. In 2014, AMR was associated with 700,000 deaths globally [6]. If the current trend of antibiotic misuse is not curtailed, about 10 million people each year will continue to die due to antibiotic resistant bacterial infections till 2050 [7]. Asia and Africa will experience the biggest loss with almost 9 million of those deaths, causing USD 210 Trillion or 7% loss of GDP (Gross Domestic Product) by 2050 [7]. Another 24 million lives will fall under extreme poverty line due to excessive out of pocket expenditure as an aftermath of AMR which will hamper Sustainable Development Goals (SDG) attainment [8].

Recent studies have established that, due to anti-microbial resistance, some mainstream infections have mutated and are requiring more than usual periods for recovery [9]. In majority of the low and middle-income countries (LMIC), irrational prescribing of antibiotics including inappropriate dosing and duration are big concerns which make the patients less compliant toward the healthcare system [10, 11]. Such unrestricted dispensing practices not only creates financial burden for the patients but also instigates them to often seek treatment for self-limiting conditions where antibiotics have no proven therapeutic benefits and can also lead to adverse health outcomes [12]. In LMICs, profound insufficiency of infection control activities, poor sanitation conditions, and poor personal hygiene maintenance also play a role in emergence of AMR [13]. In primary healthcare centers, where about 85–90% of the total antibiotic prescriptions are made, almost half of the times, the authenticity of indications of the drugs can be questioned [14]. Studies have identified that patient-pressure to prescribe antibiotics, over burden of clinical practice, over diagnosis, physicians' personal beliefs, and the lucrative monetary benefits offered from different pharmaceutical companies are key factors associated with over prescribing of antibiotics by physicians [15–20]. Knowledge gap and prevailing misconceptions are also hindering best practices of antibiotic prescribing [11]. On top of that, patients in LMICs show incomplete adherence to the treatment protocols, often not complying with the required dose and duration of specific treatment and further complicating the situation [21].

In consideration of the current scenario, it has been of paramount importance both for healthcare professionals and the general population to be more aware of anti-microbial resistance. Hence the scientific community is constantly advocating for rational prescribing of antibiotics and dispensing and consuming antibiotics at clinical settings to combat AMR threats [22]. Along with that, every individual needs to be prepared even outside the clinical settings to cope up with necessary changes in pharmaceutical management of infections [22]. In fact, the World Health Organization's (WHO) global action plan to combat AMR has emphasized effective mass communication to creating public awareness as a priority [23].

Media has historically played an important role in disseminating health messages and influence public health behavior [24, 25]. During the COVID-19 pandemic, mass media acted as the bridge between the international public health bodies (e.g., WHO and CDC) and the general people [26]. Particularly, print media can be utilized as a crucial source of knowledge to shape peoples' understanding better about science and medicine and also about potential health threats [27, 28]. In recent times, news dailies acted as gatekeepers against misinformation and disinformation during the COVID-19 outbreak [29]. Therefore, given the current upsurge of AMR, mass media can play a pivotal role in creating public awareness through disseminating tailored and targeted health information around AMR. Especially, in the current world of big pharmaceutical companies where they act as the "third dimension" of power [30] influencing the health system, the public, and the mass media, understanding the strategic narratives constructed by mass media is imperative [31, 32]. However, the impact of these pharmaceutical companies on mass media is yet to be understood holistically.

Among all the WHO global regions, currently South-East Asia has been ranked to have the highest risk of AMR crisis, according to a 2014 WHO report [5]. Being an enlisted LMIC country with a gigantic population of about 163 million, Bangladesh is currently struggling with exponential growth in cases of anti-microbial resistance (Ampicillin resistance 94.6%. Amoxiclav 67.1%, Ciprofloxacin 65.2% and Co-trimoxazole 72% for E. Coli) and is posing a great regional and global threat due to high degree of AMR [33].

Unfortunately, the role of mainstream media to create awareness about misuse of drugs and AMR remains suboptimal in the region, particularly with regard to dissemination of information around national antimicrobial policies [34]. Moreover, it needs to be understood that how the media is creating the narrative of AMR–how they are highlighting the consumer's behavior and attitude and how they are addressing the mishaps by the people in power. Therefore, we aimed to conduct an extensive search of leading national daily newspapers and identify published articles related to acknowledgement of AMR risk. The findings of this study can help identify the role print media in Bangladesh has played in disseminating information about AMR and determine the need to develop a national strategy to engage with mass media in creating mass awareness against AMR.

## Methods

### Conceptual framework

We conducted a content analysis to examine a corpus of daily printed newspaper articles in Bangladesh which are also available in the dailies' websites. For this analysis, we followed two theories–Agenda Setting Theory and News Framing Theory [35, 36]. The agenda setting theory will let us understand what news agenda has been set by the media and the news framing theory will let us understand how news articles are framed to narrate the agenda set by the media.

The media portrays issues which interest the public. In doing so, some of the contents presented and circulated by the media influence the public, and specific attributes of certain issues gets distinct response from the public. Subsequently, to keep the circulation high, the media only concentrates on those distinct topics. Thus, other issues which require more importance and assessments, get in the sidelines. That is how agendas are created [37].

In creating the agenda, framing the news holds significance of great magnitude. News framing shapes the development of a public discourse–how importance will be given to certain issues, how they will be excluded or emphasized, and how it will be communicated. News is framed keeping the social, economic, and political factors of a country [38].

Therefore, we took up these two theories to understand how the interactions of these two theories shape up the context of antimicrobial resistance (AMR) in Bangladesh. We aimed to

understand how the print media focused on certain issues, how they contributed to the gradual development of AMR narrative in Bangladesh, and which attributes of this issue got prioritized.

## Data collection

Based on the number of daily circulations, according to sources from Department of Films and Publications, Government of Bangladesh, 2018 (https://dfp.portal.gov.bd/site/page/6469f3ce-a47b-453f-84e6-c60e3732bae5/-) (Annex 1 in S1 Appendix), we purposely selected twelve highest circulated newspapers; six of which are circulated in English and six in Bangla. Usually, the financially rich people are the readers of the English dailies, and they are less in number. On the other hand, Bangla newspapers are widely read by people from all economic classes who make the biggest medicine consumer group in Bangladesh. We wanted to understand if there is a qualitative difference in reporting patterns between English and Bangla newspapers. To keep an alignment with the WHO SEARO antimicrobial resistance strategy 2010–2015 (as the strategy includes mass media and we want to see how mass media is doing in this case) [39], we included relevant articles published between January 2010 to September 2021 using the following electronic search terms for each newspaper separately, "Anti-microbial resistance", "Superbugs", "Antibiotic resistance", "Drug resistance", "AMR awareness", "AMR stewardship", "Microbial resistance", "Infectious Disease" and "Rational use of antibiotic" (Annex 2 in S1 Appendix). All the searches were done by one investigator (ZA). During this search, we did not use Boolean modifiers such as OR, AND, etc. The keywords were enough. That is because we wanted to capture all the articles that mentioned the keywords even for once. We repeated our search multiple times for cross-checking to avoid missing out on any report. Finally, we downloaded the pdf versions of all the included articles and categorized them on a database.

## Data extraction

Extracted data from each eligible article including name, and publishing date of the article along with the respective newspaper source and their web links on different columns were recorded in a spreadsheet. We also categorized all the articles into three types in accordance with their nature and content. Usually, these articles mention their types in the newspaper. Moreover, we found a similar study conducted in Austria, and followed the classification in similar way [40].

a. **Reports (news reports and feature articles)**: Articles containing information gathered or studied by the staff correspondents of the newspaper on recent events highlighting issues related to AMR or antibiotic misuse. Feature articles provide more in-depth data and insights.

b. **Opinion**: Articles where content experts expressed their opinions on AMR or antibiotic misuse either through interviews with the newspaper reporters or in their own writings [41].

c. **Editorial**: Articles constructed by the chief or sub-editors of the newspapers where they have put their thoughtful words to get an in-depth understanding of the overall scenario that cuts across multi-faceted aspects of AMR and antibiotic misuse [42].

The categorization was cross-examined by two investigators to reach an agreement over the categorization quality through an inter-observer reliability cheque.

## Data analysis

We performed a content analysis of the extracted data. We collected details of the newspapers along with their number of annual circulations, numbers of articles published in the newspapers per year, and the trend of article publication in the newspapers throughout the decade.

We extracted relevant data and summarized it from all the enlisted articles and reviewed those summaries for creating clusters with similar contents through establishing a codebook using an inductive approach. After creating codes, and going through the data several times, we divided the overall data into multiple themes and sub-themes [43] to capture all potential findings related to different aspects of AMR and antibiotic misuse. We used three spectrums to categorize the media portrayal of AMR causes and subsequent suggestions/recommendations: the consumer end (those who are receiving the AMR products), the provider end (those who are providing the AMR products), and the ecosystem end (the overall environment), and displayed them in a digital data matrix according to categories. This was done by independent coders (TH, SHI, MIH and SHK). Inter-coder reliability was 87%.

## Ethical approval

The news articles have been retrieved from the open access e-websites of the respective twelve newspaper dailies, hence it required no ethical approval.

## Results

We conducted our systematic search over the websites of twelve leading national dailies and identified 275 articles published during January 2010-September 2021. Table 1 gives an overview of the distribution of articles published by the newspapers. 'The Daily Star' published the highest number of articles (English news daily) (n = 67) covering about one fourth (24.4%) of the total articles followed by 'Prothom Alo' (Bangla daily) (n = 37, 13.5%). English dailies published more articles (58.60%Vs 41.40%) than Bangla dailies.

Fig 1 shows the trend of publication of relevant articles over the last decade. The number of articles gradually increased from 5 in 2010 to 48 in 2017, while having a sudden drop in 2020 (23) and then rising again in 2021 (41). The latter half of the decade witnessed three times publications (75%) than the earlier half. The majority (n = 184, 66.9%) of the articles published were reports, 79 (28.7%) were opinions and 12 (4.4%) were editorials.

**Table 1. Distribution of the AMR-related articles by publication in twelve national dailies.**

| Name of the Newspaper | N (%) |
|---|---|
| The Daily Star | 67 (24.4) |
| Prothom Alo | 37 (13.5) |
| Dhaka Tribune | 32 (12) |
| Jugantor | 32 (12) |
| Daily Sun | 31 (11.3) |
| Kaler Kantho | 28 (10.2) |
| The Financial Express | 14 (5.1) |
| The Daily Observer | 9 (3.3) |
| The Daily Ittefaq | 8 (2.9) |
| The Asian Age | 7 (2.5) |
| Daily Janakantha | 7 (2.5) |
| Bangladesh Pratidin | 3 (1.1) |
| **Total** | **275 (100)** |

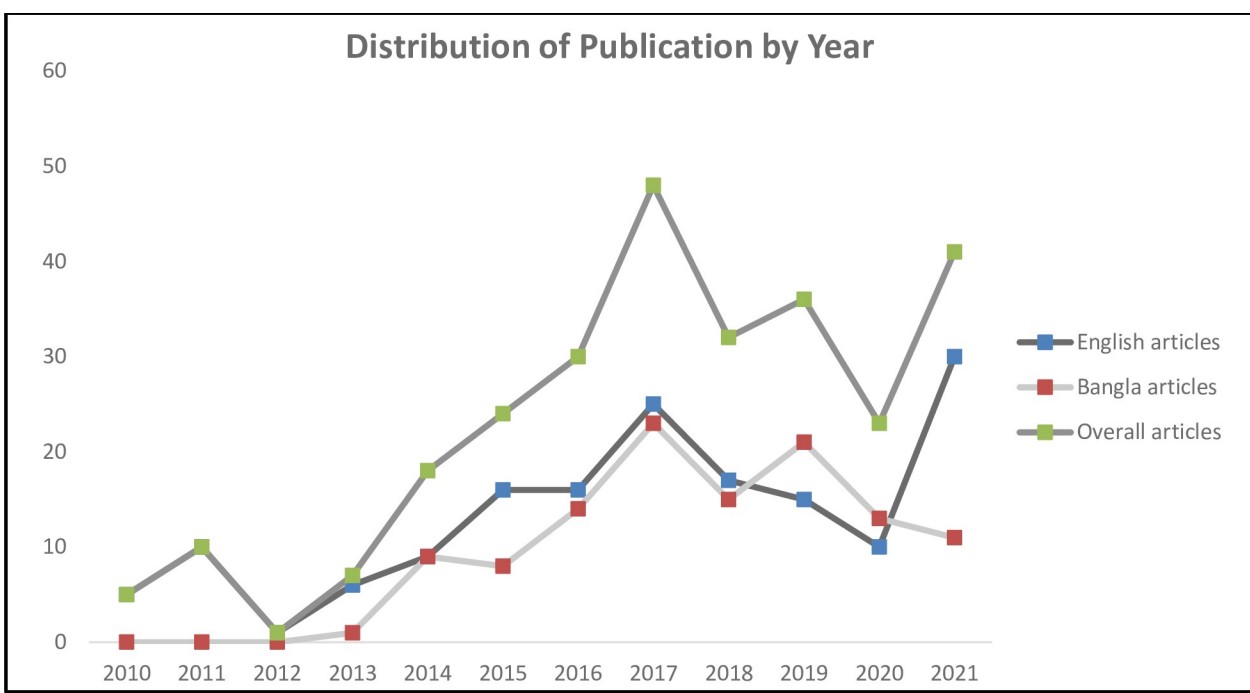

**Fig 1. Publication of articles by year.**

We also analyzed the distribution of different types of articles according to year of publication. In 2017, many of the articles were reports (n = 32) and opinions (15) whereas the highest circulation of editorials (3) was in 2021 (Fig 2).

## Thematic content findings

According to the print media, the tendency of the consumers to purchase antibiotics without a valid prescription from a qualified health professional has been the most frequently referred contributor to AMR (Table 2). Other causes reported by the print media include incomplete dosing and inappropriate duration, self-medication, and using leftover antibiotics from previous courses. In response to these causes, the reported referred recommendations were avoiding indiscriminate use of antibiotics followed by maintenance of complete dosage and duration of ongoing courses, handwashing, and hygiene maintenance, and avoiding self-medication along with left-over and adulterated antibiotics from previous prescriptions.

In terms of provider end (Table 2), the media cited two most frequent causes of AMR; widespread over the counter (OTC) antibiotic sales by drug shops in LMICs and physicians' irrational use of broad-spectrum antibiotics. According to media sources, other significant mediators to this inappropriate prescribing may be the constant pressure to meet customer demand and the absence of required and regular diagnostic facilities to prescribe specific antibiotics. The least reported causes by media were poor choice of dose, duration and root of antibiotic administration while prescribing and physicians getting lured by illegal lucrative monetary benefits offered by giant pharmaceutical companies to prioritize their products even if not required.

Regarding ecosystem end (Table 2), poor infection prevention and control (IPC), and a lack of sanitation and hygiene maintenance have been identified as the leading causes of AMR in LMIC such as Bangladesh. The prevalence of non-formal unqualified health care providers, a lack of government supervision on antibiotic sales, and excessive antibiotic promotion by

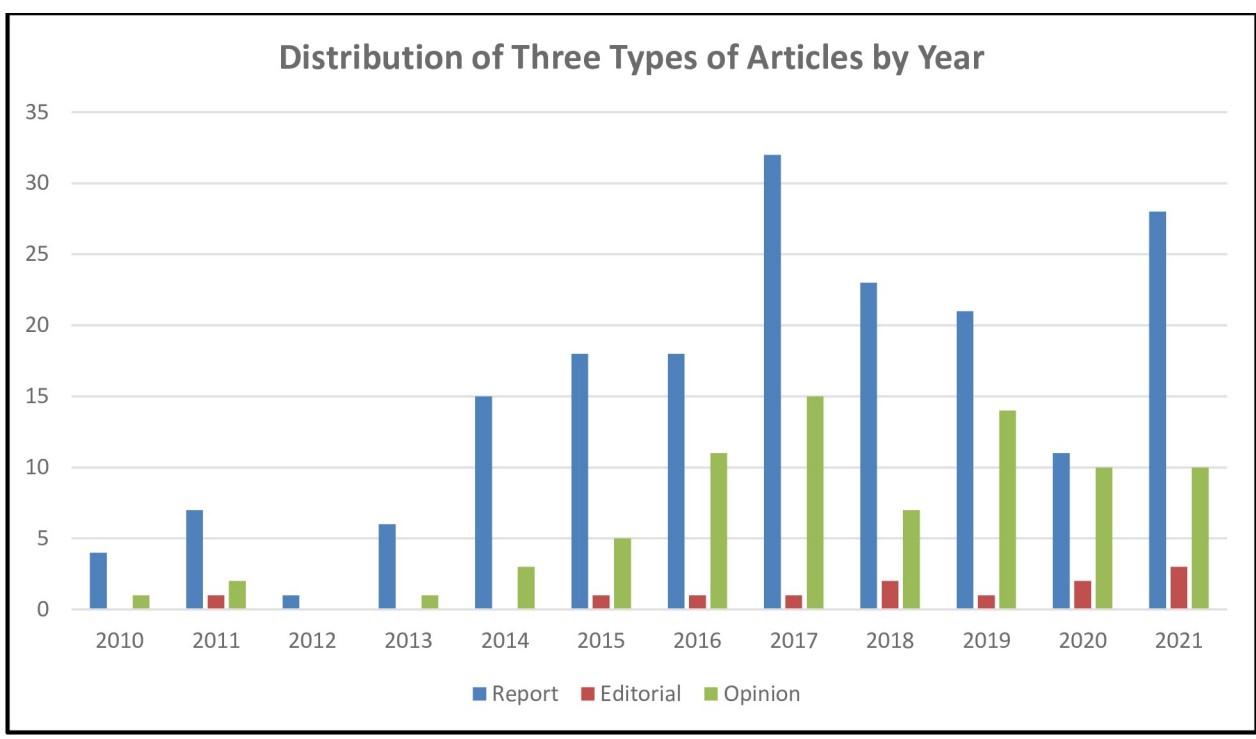

**Fig 2. Distribution of three types of articles by year.**

various pharmaceutical companies were also portrayed as ecosystem-end causes. The least reported causes were lack of national or contextualized antibiotic prescribing manuals and absence of AMR surveillance system in LMIC settings. Media has placed greater emphasis on the ecosystem end's recommendations than any of the other two spectrums. (Table 2).

The media's tendency to publish event-centric articles on AMR is depicted in Table 3. The events falling under the categories of "Bangladeshi Prime Minister's speech on a Global Leaders' Group Inauguration," "WHO Reports on elements of AMR," and "International Organization/Agency/Corporate Reports" were the main causes of the peak media attention for AMR risk. It was rarely influenced by events such as the "Bangladeshi High Court's verdict against the undisputed sale of antibiotics", "World Antibiotic Awareness Week (WAAW) observation", "MOHFW, DGHS, IEDCR, and DGDA press releases, programs, and seminars on different aspects of antibiotic use", or "Spread of superbugs in specific regions/countries" (Table 3).

## Discussion

In the context of Bangladesh, no rigorous study has been executed to identify and interpret the reporting of AMR crisis by news media and our study is one of the initial efforts in this regard. Therefore, with a focus on both English and Bangla news articles published in Bangladeshi newspapers between 2010–21, this study comprehensively analyzed how the newspaper media set the agendas of the grand narrative of AMR and framed the published news—its initiation, propagation, associated risks, solutions, and how associated stakeholders were held accountable for AMR in Bangladesh. This analysis also tried to investigate from the news framing theory that how people from different wealth classes are supplied with information and befitted in the AMR narrative through the lens of newspaper articulation and content structure.

**Table 2. Causes of AMR and adjacent recommendations narrated in the newspaper dailies across three spectrums, Bangladesh, 2010–2021.**

| | Causes | N (%) | Recommendations | N (%) |
|---|---|---|---|---|
| **Consumer End** | Purchase of antibiotics without physicians' prescription | 19 (32.2) | Avoiding unjustified or indiscriminate use of antibiotics | 8 (22.9) |
| | Patients following incomplete dose and duration of antibiotics | 17 (28.8) | Maintaining complete dose and duration of antibiotics | 8 (22.9) |
| | Self-medication of antibiotics | 16 (27.1) | Hand washing and hygiene maintenance | 7 (20) |
| | Poor adherence to antibiotic treatment | 5 (8.5) | Avoiding self-medication and use of left-over, spared, or adulterated antibiotics from previous/others prescription | 7 (20) |
| | Use of leftover antibiotics from previous courses | 2 (3.4) | Physical exercise, consumption of nutritious food, regular hydration | 5 (14.3) |
| | **Total** | **59** | **Total** | **35** |
| **Provider End** | Rampant selling of antibiotics by drug shops without prescription in LMICs | 20 (29) | Prescribing antibiotics after necessary diagnostic tests | 7 (33.3) |
| | Irrational prescribing of broad-spectrum antibiotics by physicians | 18 (26.1) | Prescribing antibiotics considering 5Rs (right patient, right drug, right dose, right route, right time) | 5 (23.8) |
| | Tension among physicians regarding loosing customer acquisition and constant pressure from patients to prescribe antibiotics | 9 (13) | Avoiding antibiotic prescription request from infodemic patients | 5 (23.8) |
| | Lack of availability of diagnostic amenities before prescribing antibiotics | 7 (10.1) | Impeding the selling of substandard quality antibiotics | 2 (9.5) |
| | Prescribing and dispensing sub-standard quality of antibiotics | 6 (8.7) | Introduction of rapid diagnostic test amenities at all tiers of health service delivery to ensure justified use of antibiotics | 2 (9.5) |
| | Choosing wrong dose, duration and route and type of antibiotic by physicians | 5 (7.2) | | |
| | Unethical prescription by physicians for lucrative monetary benefits from pharmaceutical companies | 4 (5.8) | | |
| | **Total** | **69** | **Total** | **21** |
| **Ecosystem End** | Poor IPC, sanitation, and hygiene in LMIC settings | 10 (25) | Joint and integrated multi-sectoral approach through international alliances among global leaders to fight AMR burden | 13 (14.4) |
| | Illegal prescription of antibiotics by quacks/unqualified physicians | 9 (22.5) | More investment from pharmaceutical giants in AMR research and formulation of new and highly efficacious antibiotics | 13 (14.4) |
| | Lack of government monitoring on antibiotic sales | 8 (20) | Strict law enforcement (including penalty and punishment system) against OTC antibiotic selling | 12 (13.3) |
| | Drastic promotion of antibiotics by pharmaceutical companies | 8 (20) | Functionalizing and strengthening AMR surveillance system | 11 (12.2) |
| | Lack of national guidelines/instruction manuals on the use of antibiotics | 2 (5) | Robust media campaigns to raise awareness against AMR | 10 (11.1) |
| | Geographic unequal distribution of newer efficacious antibiotics | 2 (5) | Formulating and strengthening guidelines on antibiotic use and adjacent policies at national level | 8 (8.9) |
| | Poor AMR Surveillance System | 1 (2.5) | Strict government monitoring to revoke the rampant selling of antibiotics without prescription | 7 (7.8) |
| | | | Ensuring proper IPC, WASH, and waste management in LMIC healthcare settings | 6 (6.7) |
| | | | Ensuring equitable geographic access to high quality antibiotics through international collaborations | 6 (6.7) |
| | | | Research capacity development in Bangladesh regarding AMR | 2 (2.2) |
| | | | Registration of unregistered pharmacies and antibiotic taxation | 2 (2.2) |
| | **Total** | **40** | **Total** | **90** |

The role of media (television, digital and printed news) in awareness building among people about AMR and the quality of such risk reporting has been widely investigated in other countries including Australia, North America, and Germany [43–45]. The investigations also attempted to examine the pattern and meaning of AMR related news in these countries. Most of these studies suggested that AMR crisis had been a fragmented story principally framed by

**Table 3. Notable events that triggered peak media attention to AMR in Bangladesh, 2010–2021.**

| Article on Prominent Event | Total number N (%) |
|---|---|
| Prime Minister Sheikh Hasina's speech as a co-chair on One Health Global Leaders' Group Inauguration | 28 (22.2) |
| WHO Reports and Press Releases on various aspects of AMR | 25 (19.8) |
| International Organization/Agency/Corporate Reports (Public Health Foundation England, European Centre for Disease Prevention and Control, Global review on AMR– 2014, Organization for Economic Co-operation and Development, Public Health Department England, Global and foreign policies) | 18 (14.3) |
| Other events (trainings and seminars by non-government national/international organizations and Universities) | 12 (9.5) |
| Bangabandhu Sheikh Mujib Medical University (BSMMU) and International Centre for Diarrheal Disease Research, Bangladesh (icddr,b) study results | 10 (7.9) |
| Bangladeshi High Court's verdict against undisputed selling of antibiotics in pharmacies | 5 (3.4) |
| Observation of World Antimicrobial Awareness Week | 5 (3.4) |
| Ministry of Health and Family Welfares (MOHFW), Directorate General of Health Services (DGHS), Institute of Epidemiology, Disease Control and Research (IEDCR), and Directorate General of Drug Administration (DGDA) press releases, programs, and seminars on various aspects of antibiotic use | 5 (3.4) |
| Spread of superbugs in certain regions/countries | 4 (3.2) |
| Different International University/Organization study results | 32 (25.4) |
| **Total** | **126** |

scientific discovery and had restricted appeal mostly to the social standings of mainstream researchers. On top of that, the studies identified a lack of approaches by the media houses to contextualize the risk information and sticking to absolute numbers which makes it challenging for audience/readers to decipher information. Although most of the news events studied in these investigations were termed to be "well-informed, accurate, balanced and responsive to the issue at stake," still the extent to which they had an actual impact on improving public understanding and awareness around AMR crisis remained unascertained.

In our study, we conducted a content analysis of 275 news articles published in twelve leading Bangladeshi newspaper dailies between 2010–2021. Over our study period, it was evident that both Bangla and English articles number were gradually rising, as 75% of the news reports were published in later half of the study period (2017–2021), which is in accordance with the similar studies conducted in China and Germany that reported an increasing trend in Media attention for AMR between 2015–2017 particularly in less developed countries [45, 46]. However, a sudden drop of frequency in 2020 might be possible due to the emergence of COVID-19 pandemic which globally was the main news item throughout 2020 and continues to be a major component of news articles.

According to the agenda setting theory, the occurrence of the COVID-19 pandemic compelled the media houses publish more news about the pandemic because the public demanded it, and as a result AMR news reports got sidelined. This was indeed beneficial to the public. The mass media played a huge role in preventing an infodemic (overwhelming quantity of information including incorrect and misleading information) filled with disinformation and misinformation regarding the pandemic, subsequent vaccination campaigns, and treatment procedures [47].

Our results suggest while reporting has increased over the years, the number of reports remain significantly low (27 per year, around 2 articles per month), which shows a large ignorance of newspaper media to the topic of AMR, despite it being a leading global health issue and media being a crucial stakeholder to fight back against its burden. Moreover, English dailies published

more AMR news than Bangla newspapers which might have caused a disproportionate sharing of information among the population since Bangla is the national language and English articles are still not widely available, particularly in sub-urban and rural communities. Moreover, it indirectly indicates that the people who are poor and at the same time, the principal consumers of medicines, receive a lower coverage of awareness news than the rich people. In an article by Sánchez & Sivaraman conducted in India, it was identified that low priority was given to scientific and health journalism where political and economic news were of top priorities [48]. Moreover, from the lens of the agenda setting theory, we see that economic class decisively determines which news to be set for a particular set of readers. In the context of Bangladesh, the rich people usually subscribe to the English dailies who hold lesser proportion in the political context [49]. Therefore, agendas for people who do not belong to rich societies, remain distinctively different.

We observed compelling misalignments and the use of complicated jargons in many instances. This may be linked with poorer understanding of the scientific topics among the journalists. From the lens of news framing, jargons make an issue serious but takes away the opportunity for understanding.

The highest reported cause of AMR regarding 'consumer end' was consumption of antibiotics without physicians' prescription which was consistent with findings from study done by Chen et al (2018), which identified retail pharmacies' easy access to antibiotics without a prescription as the primary cause of AMR worldwide [50]. Although the recommended approach by most articles to combat this was 'declining the unjustified use of antibiotics,' most of the reports lacked detailed explanations on ways on reducing the misuse.

Besides, two significant causes of AMR in terms of 'provider end' were the least reported causes across all news articles and they are–the constant patient pressure upon physicians to prescribe antibiotics for attaining patient satisfaction and receiving bribe/monetary incentives from the pharmaceutical companies to promote their products. Though these factors play a crucial role in contributing to AMR surge [11, 51–53], somehow, the national dailies did not cover them. This finding needs more emphasis as it is indicative of control from the big pharmaceutical companies over the media. In Bangladesh, many of the media houses are owned by the parent companies of the pharmaceutical companies. This may have a severe impact on free and transparent journalism. However, this phenomenon needs more investigation.

The least reported recommendation for 'provider end' by the news articles was availability of rapid diagnostic tests to differentiate between viral and bacterial causes of illness in health facilities at different levels. This can contribute significantly to reducing inappropriate use of antibiotics and be a cost-friendly and impactful initiative to help ensuring rational antibiotic prescribing [54, 55]. Also, we did not identify any recommendation to take legal measures against OTC selling of antibiotics without asking for valid prescriptions although this was the most frequently reported cause of AMR regarding 'provider end.'

We also noticed discrepancy in recommendations to prevent AMR under 'ecosystem end' as there were no suggestions from the news media for undertaking stringent measures against pharmaceutical companies for aggressive promotion of their products among physicians. They used non-comprehensive and sugar-coated terms and jargon. For example, the articles did not describe how joint and integrated multi-sectoral approach through international alliances could curb the global burden of AMR. Moreover, practical suggestions like licensing of unregistered pharmacies in Bangladesh or enhancing research capacity of Bangladesh to detect AMR patterns in local context were least emphasized.

We found that the greatest number of reports published were covering the events where the Prime Minister of Bangladesh was associated. Previous studies [28, 46] have also reported that official assessment reports of AMR hazards may lead to increase in media interest, as government agencies, healthcare professionals and scientists are considered to be the major social

actors in communicating AMR. These studies have also suggested that practical and promotional events that may create awareness around AMR, such as observance of world antibiotic awareness week (WAAW), were not covered by newspapers with great enthusiasm. This can be related to a tendency to publish only prominent event centric articles which increase the circulation of the newspapers and target rating point (TRP) of news media. Nonetheless, the government needs to show the political commitment to prevent AMR by taking more initiatives like public events, thus gaining more media coverage. Additionally, the media should develop a bridge with the scientific community and highlight more AMR research studies so that the public can be aware of the recent developments in the field.

There were several distinct features of Bangladeshi news articles too. There were few investigative feature reports which only connected with the lived experiences of patients, and in some case, the health providers. There were some dramatic uses of jargons but the impact of such jargons is out of scope of this study. However, the more jargons are avoided, the better is the understanding for the mass people.

Another aspect of AMR reporting can be to bring more personal experiences. It is necessary to portray more stories with which people can connect. Just to understand how people engage with news reports, we accessed some of the news reports on their respective Facebook page. We observed feature reports from the Bangla dailies had more reads because of their humane touch. This kind of reporting where the struggle of AMR is portrayed, can contribute more to AMR awareness, rather than formulating a blame game. Though checking the online engagement was not a part of the study methodology, we recommend checking this engagement to conduct digital ethnography. Also, understanding and reviewing the engagement on social media, keeping the two theories discussed in the paper, we will be able to understand how people interact with public health messages. It will help public health researchers and the government to formulate effective communication with the public.

Another aspect to improve the quality of AMR reporting is to bring in new thoughts such as one-health. Currently, one of the main reasons of AMR in Bangladesh is the use of antibiotics in poultry feed [56]. The one-health approach will instigate people to think about the environment as a whole, and may contribute to preaching of a holistic understanding on AMR.

Now, let us dissect the whole scenario using the two theories. At first, the agenda setting theory tells us that the agenda of AMR only gets priority when people in power are associated. That is why more news was published centering around a national event, or if influential people such as the Prime Minister visited any international event regarding AMR. This succinctly shows that AMR is only an agenda for the media, not for the public. Now, after the 2014 WHO report on AMR, we see an increase in AMR-related publication (Fig 1). Therefore, we can predict that there was a drive for setting the agenda of raising awareness among the public on AMR as international bodies were associated. However, due to lack of scientific journalism, we did not have sufficient investigative journalism in this regard. On the contrary, we only had articles which framed the consumers and the patients to be responsible for increasing AMR. The news headlines and the news bodies were framed as such–only blaming the consumers, or in some cases, the providers. However, there was a heavy dearth of news articles where the role of giant corporations was reported in-depth. Though we know people buy antibiotics however and whenever they like, we do not know the reasons why they are being sold without valid prescriptions from registered doctors. Also, we have not seen articles stating why the doctors prescribe unnecessary antibiotics. We only got news framed up to blaming the primary providers, and consumers. It should be noted that without stating the ecosystem of the antibiotic market and regulations, it is futile to raise AMR awareness. Recently, the Government of Bangladesh has drafted a policy to ban antibiotic selling without prescriptions. However, how this decision will be enacted still larks in dark.

This study confronts several limitations. Firstly, our searches were limited to Bangla and English articles of only the widely circulated twelve newspapers. Not all news media publishing online, and offline articles were included in this study. Hence, we might have missed reports from other newspapers which turns the reported article number into an underestimation. However, the included newspaper in our search has a wider readership of around 2.5 million which covers a substantial portion of daily newspaper readers in Bangladesh. Secondly, the retrieval of news articles was done retrospectively, which makes the articles subject to being lost over time as websites of majority of the Bangladeshi newspapers refresh their repository in frequent intervals (thrice in last 10 years). We have therefore used rigorous search keywords to maximize our search radius so that the number of lost articles becomes minimal.

Despite the limitations, our finding clearly showed significant under-reporting of AMR, a pressing global health problem and highlighted the need for more awareness among the media community on this issue. Our analysis was limited to print media and further research should focus on television and social media to better understand the breadth of the coverage and to explore how such news coverage can be improved to enhance public understanding and awareness on AMR.

## Conclusion

AMR news coverage can play a role in creating mass awareness, making providers accountable and support Bangladesh' national action plan in mitigating AMR threat. The key factor to influence the greater AMR news coverage depends upon the availability, consistency, and acceptability of the articles with better explanation and analysis from the journalists. The Bangladeshi journalists who are interested in reporting AMR issues should focus on disseminating more Bangla articles, gathering scientific information, and reporting causes and recommendations responsibly. Addressing the growing need for health journalism through in-house capacity building training may help in more sensible and appropriate reporting around emerging public health issues including AMR.

## Supporting information

**S1 Appendix.**
(DOCX)

**S1 File.**
(XLSX)

## Acknowledgments

The authors gratefully thank the authorities of leading newspaper dailies for open access news articles on their respective websites. The authors are also very much thankful to the core donors of the International Centre for Diarrheal Disease Research, Bangladesh (icddr,b) for their generous ongoing support provided to various research projects. icddr,b acknowledges with gratitude the commitment of the governments of Bangladesh, Canada, Sweden, and the UK for providing unrestricted support.

## Author Contributions

**Conceptualization:** Tahmidul Haque, Syed Hassan Imtiaz, Md. Imran Hossain, Sazzad Hossain Khan, Md. Mahfuj Alam, Zahidul Alam, S. M. Rokonuzzaman, Orindom Shing Pulock, Nusrat Homaira, Md. Zakiul Hassan.

**Data curation:** Tahmidul Haque, Sazzad Hossain Khan, Haroon Bin Murshid.

**Formal analysis:** Tahmidul Haque, Syed Hassan Imtiaz, Md. Imran Hossain, Sazzad Hossain Khan, Md. Mahfuj Alam.

**Investigation:** Tahmidul Haque, Syed Hassan Imtiaz, Md. Imran Hossain, Sazzad Hossain Khan, Md. Mahfuj Alam, Nusrat Homaira, Md. Zakiul Hassan.

**Methodology:** Tahmidul Haque, Syed Hassan Imtiaz, Md. Imran Hossain, Sazzad Hossain Khan, Nusrat Homaira, Md. Zakiul Hassan.

**Project administration:** Tahmidul Haque.

**Resources:** Tahmidul Haque, Syed Hassan Imtiaz, Sazzad Hossain Khan.

**Supervision:** Nusrat Homaira, Md. Zakiul Hassan.

**Validation:** Tahmidul Haque, Syed Hassan Imtiaz.

**Visualization:** Tahmidul Haque, Syed Hassan Imtiaz, Md. Imran Hossain, Sazzad Hossain Khan, Md. Mahfuj Alam, Nusrat Homaira, Md. Zakiul Hassan.

**Writing – original draft:** Tahmidul Haque, Syed Hassan Imtiaz, Susmita Dey Pinky.

**Writing – review & editing:** Ataul Karim Arbi, Nusrat Homaira, Md. Zakiul Hassan.

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
