## [Decision Letter · Decision Letter 0]

13 Oct 2023

PONE-D-23-26308Antimicrobial Resistance in Bangladeshi Newspapers During 2010-2021: Discussions Around the CrisisPLOS ONE

Dear Dr. Haque,

Thank you for submitting your manuscript to PLOS ONE. After careful consideration, we feel that it has merit but does not fully meet PLOS ONE’s publication criteria as it currently stands. Therefore, we invite you to submit a revised version of the manuscript that addresses the points raised during the review process.

ACADEMIC EDITOR:

Experrts believe that your manuscript needs improvement overall, and their comments are mentioned in this email which will help you shape up this research in a better manner.

We look forward to receiving your revised manuscript.

Kind regards,

Muhammad Farooq Umer, PhD Epidemiology and Health Statistics

Academic Editor

PLOS ONE

Journal Requirements:

2.  Thank you for submitting the above manuscript to PLOS ONE. During our internal evaluation of the manuscript, we found significant text overlap between your submission and previous work in the [introduction, conclusion, etc.].

Please revise the manuscript to rephrase the duplicated text, cite your sources, and provide details as to how the current manuscript advances on previous work. Please note that further consideration is dependent on the submission of a manuscript that addresses these concerns about the overlap in text with published work.

[If the overlap is with the authors’ own works: Moreover, upon submission, authors must confirm that the manuscript, or any related manuscript, is not currently under consideration or accepted elsewhere. If related work has been submitted to PLOS ONE or elsewhere, authors must include a copy with the submitted article. Reviewers will be asked to comment on the overlap between related submissions (http://journals.plos.org/plosone/s/submission-guidelines#loc-related-manuscripts).]

We will carefully review your manuscript upon resubmission and further consideration of the manuscript is dependent on the text overlap being addressed in full. Please ensure that your revision is thorough as failure to address the concerns to our satisfaction may result in your submission not being considered further.

Reviewers' comments:

Reviewer's Responses to Questions

**Comments to the Author**

1. Is the manuscript technically sound, and do the data support the conclusions?

Reviewer #1: Yes

Reviewer #2: Yes

2. Has the statistical analysis been performed appropriately and rigorously? 

Reviewer #1: Yes

Reviewer #2: Yes

3. Have the authors made all data underlying the findings in their manuscript fully available?

Reviewer #1: No

Reviewer #2: Yes

4. Is the manuscript presented in an intelligible fashion and written in standard English?

Reviewer #1: Yes

Reviewer #2: Yes

5. Review Comments to the Author

Reviewer #1: Th authors have examined news stories on antimicrobial resistance in Bangladeshi newspapers published between to understand the narrative on the issues from the context of a developing country like Bangladesh. This is an interesting, time and relevant topic to investigated. It has high potential to contribute to the fields of mass media role in public health communication.

Title: The second part of the title of the article confusing. I recommend to write the title as follows: “The Portrayal of Antimicrobial Resistance in Bangladeshi Newspapers During 2010-2021: Toward understanding the narrative”

Abstract: The abstract does communicate the results of the study properly. I recommend to give deeper reflection of the results in the abstract

Introduction: The background should incorporate all aspect to the topic being investigated. But this paper lacks literature on media, its role in society in forming public opinion, raising awareness, it role in public health communication.

Methods: The method is not theoretically grounded. I recommend to use “Media Framing Theory” and “Agenda Setting Theory” to in extracting data, coding and analyzing the data.

Results: I found the results are communicated properly. But its improvement using more deeper and wider lenses of mass media theories and methods

Discussion: In this part, the authors should add more mass media related literatures to build the uniqueness of their findings

Conclusion: It seems good. But needs improvement

Reviewer #2: This paper provides an analysis of digitised print media reporting on AMR in Bangladesh. It takes data from 12 newspapers (in 2 languages: Bangla and English) and between 2010 and 2021. This gives the work excellent scope. The paper categories the type of reporting and makes several interesting findings. I believe the work is methodologically sound, and well-articulated: the article is well written and structured logically. I agree with the authors that no such study has been conducted in this context, and that this makes the research timely and valuable. As the authors also note in their discussion, it is vital to keep talking about AMR, and not have the problem overshadowed by COVID-19 media reporting. This paper is a valuable contribution in that regard. I offer a few suggestions and comments below, as minor revisions before publication. These largely related to 1) minor clarifications 2) amplification of important points.

Background section:

The point about ‘big pharma’ and media power (page 5; line 85) is a little weak, and the referenced material is only generally applicable. I suggest the authors integrate the point with local Bangladeshi context. For example, is direct-to-consumer drug advertising legal? And therefore, does it pose a smaller/larger issue than other nations? It is implied (page 20) that OTC is legal, so state this sooner.

Method section:

A comment may be needed in the data collection section, about the use (or not) of modifiers in search. For example, Boolean search (OR, AND, *) etc. Even a comment on whether these were possible on the newspaper websites or not, would be valuable. If they were possible, terms like ‘superbugs’ in plural may have been better served by superbug*. These practices are common details of media research methods and might be important to note.

Discussion points:

‘One Health’ is mentioned in relation to a speech (page 16) big is not discussed in the paper as an overall strategy. One health is becoming a popular framework for many: both in terms of writing about pharmacy operations (in humans and in industries like livestock), and also writing about media reporting and how AMR is framed. It may be useful to comment in this brief inclusion of One Health and how it differs from other regions.

As well as the paper commenting on how Bangladesh might form a national strategy, the authors might like to also comment on what other nations can learn from the media reporting in Bangladesh. The matter of ‘over prescription’ is a universal topic in AMR media, but other areas seem more unique… It seems very significant that there is reporting on the ecosystem level of the problem! In other studies of AMR media, these complex issues are often overlooked in media. As the authors note, ‘scientific discovery’ based reporting is very common. The authors could note this difference and the significance.

As the authors also note, ‘event’ based reporting is common. Additional connections and difference could be made here. There is research on media (including social media) and various ‘antibiotic awareness’ initiatives. Often these end up only reaching medical professionals and not impacting the community. So, it seems significant that Bangladeshi leaders are addressing AMR awareness in the media, and this is a point of difference from what is seen in international research.

As a final connection to existing work: while the authors do not need to explore linguistics (that is not the approach of the paper), it is notable that superbugs are not discussed much in the paper. Existing research suggests that superbugs are reported on in very dramatic ways and focus on individual suffering. The appearance (or not) of they may be worth commenting on. The main point is where a victim – for example, of MRSA infection – is situated. They are not a consumer, but they suffer from the ramifications of AMR. Is this a feature of Bangladeshi media reporting? Do the authors perhaps see this type of story as part of the ‘provider end’ (the result of irrational prescription is the development of drug-resistance). Connections can be made here to Bouchoucha et al. (2019) Davis et al. (2020) and Capurro (2020) – all already cited in the paper.

Minor line notes:

Abstract: ‘disbursed’ seems intended to be ‘dispersed’

Hyperlinks: sources that are present only as hyperlinks may need to be formatted correctly, depending on the journal’s preference.

Line 87: ‘in a while’ – this needs to be clarified. Has SEA been evaluated to be currently at high risk, or is it consistently high risk?

Line 112-3: what is the ‘alignment’ with the WHO strategy? Is this just the date connection of 2010, or are the keywords shaped by the strategy?

Line 126-36: an academic methodological source may be preferable to an online BBC article, to demonstrate a systematic approach to the differences between article types.

Line 139: ‘year wise numbers’ is a little unclear

Line 193: ‘sells’ should be ‘sales’

Line 251: language could be refined around the explanation of the study in India. 'Done in' is a little informal.

6. PLOS authors have the option to publish the peer review history of their article (what does this mean?). If published, this will include your full peer review and any attached files.

Reviewer #1: **Yes: **Mohammad Aminul Islam

Reviewer #2: No

---

## [Author Response · Author response to Decision Letter 0]

26 Nov 2023

A rebuttal letter has been uploaded in the 'Attach Files' section, labeled as 'Response to reviewers', that contains all the responses to specific reviewer and editor comments.

---

## [Decision Letter · Decision Letter 1]

14 Mar 2024

PONE-D-23-26308R1The Portrayal of Antimicrobial Resistance in Bangladeshi Newspapers During 2010-2021: Toward understanding the narrativePLOS ONE

Dear Dr. Haque,

Thank you for submitting your manuscript to PLOS ONE. After careful consideration, we feel that it has merit but does not fully meet PLOS ONE’s publication criteria as it currently stands. Therefore, we invite you to submit a revised version of the manuscript that addresses the points raised during the review process.

We look forward to receiving your revised manuscript.

Kind regards,

Muhammad Farooq Umer, PhD Epidemiology and Health Statistics

Academic Editor

PLOS ONE

**Additional Editor Comments:**

The manuscript needs major changes throughout. Detailed comments from the reviewer are attached. 

Reviewers' comments:

Reviewer's Responses to Questions

**Comments to the Author**

1. If the authors have adequately addressed your comments raised in a previous round of review and you feel that this manuscript is now acceptable for publication, you may indicate that here to bypass the “Comments to the Author” section, enter your conflict of interest statement in the “Confidential to Editor” section, and submit your "Accept" recommendation.

Reviewer #2: (No Response)

Reviewer #3: (No Response)

2. Is the manuscript technically sound, and do the data support the conclusions?

Reviewer #2: Partly

Reviewer #3: Yes

3. Has the statistical analysis been performed appropriately and rigorously? 

Reviewer #2: N/A

Reviewer #3: Yes

4. Have the authors made all data underlying the findings in their manuscript fully available?

Reviewer #2: Yes

Reviewer #3: No

5. Is the manuscript presented in an intelligible fashion and written in standard English?

Reviewer #2: Yes

Reviewer #3: Yes

6. Review Comments to the Author

Reviewer #2: As in the initial version, this manuscript presents analysis of media reporting on AMR in Bangladesh (2010-2021) and makes relevant and important contributions to this area of studies. For the R1 manuscript, the additions made about big pharma (page 6/22) media ownership (page 22) and One Health (page 24) are important and add to the paper’s interesting insights. I still believe that the work is valuable to scholars of health and media.

R1 however, introduces some new issues in terms of how initial reviewer comments are addressed. In short, I do not see how the proposed theoretical frameworks are integrated and do not believe they have been appropriately reflected in the work. The manuscript therefore requires further revisions. I offer more detailed comment below:

Method:

The changes made to the Method section following review, are lacking detail. I note that the additional sentences (lines 136-140) explain the theories with nothing more than the name of the theory: ‘agenda setting theory will let us understand what news agenda has been set’ / ‘news framing theory will let us understand how news articles are framed’. This is too simple and does not show much engagement with either the theory or methods related to the theories. I believe the authors also need to think about the connection between the social psychology elements of such approaches and the media/content base of the paper, since these are quite different!

This is all quite necessary for two reasons: 1) If comments from the initial review about these theories is really being actioned, there should be deeper reflection here that is met by changes in the paper. 2) A broad journal like PLOS One has a very diverse readership, and the theories need to be explained in more detail.

I am also not sure that the single sources cited are the best choices for a clear picture of what exactly the theories do in the paper. The agenda setting article cited repeatedly is a conversation piece that relies on many other works by McCombs. The article refers to a book called ‘Setting the Agenda’ – perhaps this is a better source to reference? I think the authors need to reflect more on if/how these theories can actually add something to the paper, rather than just responding to the reviewer suggestions.

I also maintain that the authors need to refining reference choices (at line 163-4). I agree with the authors comments that resources like the BBC are quality journalism. But the sources cited (including the CSUSM Library guide now added) are student-facing resources for media type identification; they do not then flow on detail use of the types for analytical research purposes. Other academic publications that discuss this (or similar) classification methods (and their relative importance) for media articles are necessary. I think this is very important, because new insights in the discussion about feature articles, jargon etc. (lines 362-371) give weight to distinctions between article types. Since this makes valuable about how information about AMR communication differs across these article types, the support of existing academic methods is needed.

Discussion:

Since the new theories are being mentioned in the paper, they need to be more robustly reflected in the findings. ‘Agenda setting’ appears minimally (page 21) in reference to COVID. Which is somewhat prosaic and rushed… ‘Infodemic’ is not defined (again remember PLOS readership is diverse) and I am unsure if has been entirely prevented by mass media… Regardless, how does this connect back to AMR, which is already not often reported on?

Points about the status of readership (wealth) are also light on detail. What evidence is there – beyond intuition – that the economic class of a newspaper’s journalists and audiences equates to their education level? This work analysed the content of AMR reporting, not the readership demographics/understandings. These points then, seem more like future directions.

Points about personal stories seem important to consider, but the evidence here – ‘more reads’ (line 369) – is not referred to elsewhere in the paper. Did the authors have access to the publication metrics? If so, this needs to be described as part of the method. The authors also miss the opportunity to connect back to literature here. In sources cited already, the subject of human stories is mentioned. Linking back to this would aid the arguments made.

Line notes:

Line 106: why is the US based CDC being referred to as an international health body?

Line 117: sentence on WHO’s evaluation of AMR risk in South-East Asia remains confusing. Source date should be mentioned in text. Source it 10 years old, so cannot be seen as ‘recent’. What, definitionally, is a crisis?

Line 125-7: since the paper does not share research questions or hypothesis motivating the study, I would caution against the use of this kind of rhetorical question.

Line 148-9: point about ‘alignment’ with the WHO SEARO strategy remain unclear. Please clarify, for example, that the keyworks are a feature of the strategy.

Table 2: following the changes elsewhere, ‘sells’ should be ‘sales’

Line 330-4: very long sentence with confusing conjunctions. Requires revising.

Line 340-1: there is quoted text from media here, please add reference.

Reviewer #3: (No Response)

7. PLOS authors have the option to publish the peer review history of their article (what does this mean?). If published, this will include your full peer review and any attached files.

Reviewer #2: No

Reviewer #3: **Yes: **Martha F Mushi

---

## [Author Response · Author response to Decision Letter 1]

28 Apr 2024

All the comments have been reviewed and attached as a rebuttal letter named "Response to Reviewers".

---

## [Editor Report · Decision Letter 2]

15 May 2024

The Portrayal of Antimicrobial Resistance in Bangladeshi Newspapers During 2010-2021: Toward understanding the narrative

PONE-D-23-26308R2

Dear Dr. Haque,

We’re pleased to inform you that your manuscript has been judged scientifically suitable for publication and will be formally accepted for publication once it meets all outstanding technical requirements.

Kind regards,

Muhammad Farooq Umer, PhD Epidemiology and Health Statistics

Academic Editor

PLOS ONE
---

## [Editor Report · Acceptance letter]

22 May 2024

PONE-D-23-26308R2 

PLOS ONE

Dear Dr. Haque, 

I'm pleased to inform you that your manuscript has been deemed suitable for publication in PLOS ONE. Congratulations! Your manuscript is now being handed over to our production team.

Kind regards, 

on behalf of

Dr. Muhammad Farooq Umer 

Academic Editor

PLOS ONE